# Scalable Graph Neural Networks for Heterogeneous Graphs

## Abstract

Graph neural networks (GNNs) are a popular class of parametric model for learning over graph-structured data. Recent work has argued that GNNs primarily use the graph for feature smoothing, and have shown competitive results on benchmark tasks by simply operating on graph-smoothed node features, rather than using end-to-end learned feature hierarchies that are challenging to scale to large graphs. In this work, we ask whether these results can be extended to *heterogeneous* graphs, which encode multiple types of relationship between different entities. We propose *Neighbor Averaging over Relation Subgraphs* (NARS), which trains a classifier on neighbor-averaged features for randomly-sampled subgraphs of the "metagraph" of relations. We describe optimizations to allow these sets of node features to be computed in a memory-efficient way, both at training and inference time. NARS achieves a new state of the art accuracy on several benchmark datasets, outperforming more expensive GNN-based methods.

## 1 Introduction

In recent years, deep learning on graphs has attracted a great deal of interest, with new applications ranging from social networks and recommender systems, to biomedicine, scene understanding, and modeling of physics (Wu et al., 2020). One popular branch of graph learning is based on the idea of stacking learned "graph convolutional" layers that perform feature transformation and neighbor aggregation (Kipf & Welling, 2017), and has led to an explosion of variants collectively referred to as Graph Neural Networks (GNNs) (Hamilton et al., 2017; Xu et al., 2018; Velickovic et al., 2018). Most benchmarks for learning on graphs focus on very small graphs, but the relevance of such models to large-scale social network and e-commerce datasets was quickly recognized (Ying et al., 2018). Since the computational cost of training and inference on GNNs scales poorly to large graphs, a number of sampling approaches have been proposed that improve the time and memory cost of GNNs by operating on subsets of graph nodes or edges (Hamilton et al., 2017; Chen et al., 2017; Zou et al., 2019; Zeng et al., 2019; Chiang et al., 2019).

Recently several papers have argued that on a range of benchmark tasks – social network and e-commerce tasks in particular – GNNs primarily derive their benefits from performing feature smoothing over graph neighborhoods rather than learning non-linear hierarchies of features as implied by the analogy to CNNs (Wu et al., 2019; NT & Maehara, 2019; Chen et al., 2019; Rossi et al., 2020). Surprisingly, Rossi et al. (2020) demonstrate that a one-layer MLP operating on concatenated N-hop averaged features, which they call Scalable Inception Graph Network (SIGN), performs competitively with state-of-the-art GNNs on large web datasets while being more scalable and simpler to use than sampling approaches. Neighbor-averaged features can be precomputed, reducing GNN training and inference to a standard classification task.

However, in practice the large graphs used in web-scale classification problems are often heterogeneous, encoding many types of relationship between different entities (Lerer et al., 2019). While GNNs extend naturally to these multi-relation graphs (Schlichtkrull et al., 2018) and specialized methods further improve the state-of-the-art on them (Hu et al., 2020b; Wang et al., 2019b), it is not clear how to extend neighbor-averaging approaches like SIGN to these graphs.

In this work, we investigate whether neighbor-averaging approaches can be applied to heterogeneous graphs (HGs). We propose *Neighbor Averaging over Relation Subgraphs* (NARS), which computes neighbor averaged features for random subsets of relation types, and combines them into a single

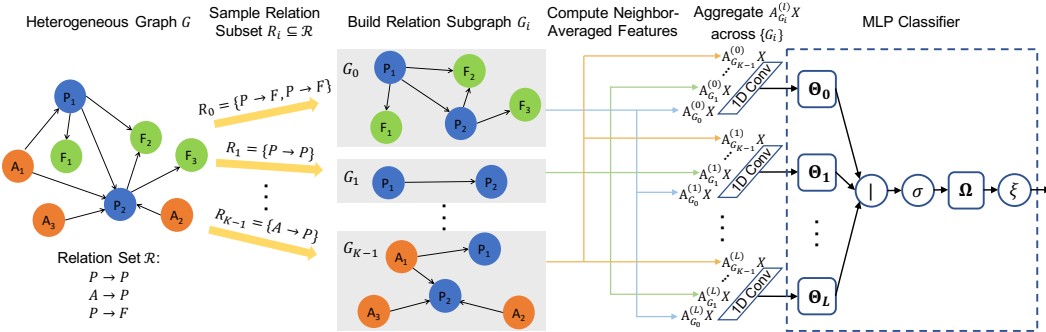

Figure 1: Neighbor Averaging over Relation Subgraphs on heterogeneous graph $G$. $G$ has three node types: Paper (P), Author (A), and Field (F), and three relation types: Paper cites Paper (P→ P), Paper belongs-to Field (P→F), Author writes Paper (A→P).

set of features for a classifier using a 1D convolution. We find that this scalable approach exceeds the accuracy of state-of-the-art GNN methods for heterogeneous graphs on tasks in three benchmark datasets.

One downside of NARS is that it requires a large amount of memory to store node features for many random subgraphs. We describe an approximate version that fixes the memory scaling issue, and show that it does not degrade accuracy on benchmark tasks.

## 2 BACKGROUND

Graph Neural Networks are a type of neural model for graph data that uses graph structure to transform input node features into a more predictive representation for a supervised task.

A popular flavor of graph neural network consists of stacked layers of operators composed of learned transformations and neighbor aggregation. These "message-passing" GNNs were inspired by spectral notions of graph convolution (Bruna et al., 2014; Defferrard et al., 2016; Kipf & Welling, 2017). Consider a graph $G$ with $n$ vertices and adjacency matrix $A \in R^{n \times n}$. A graph convolution $g \star x$ of node features $x$ by a filter $g$ is defined as a multiplication by $g$ in the graph Fourier basis, just as a standard convolution is a multiplication in Fourier space. The Fourier basis for a graph is defined as the eigenvectors $U$ of the normalized Laplacian, and can be thought of as a basis of functions of varying smoothness over the graph.

$$g \star x = U g U^T x \tag{1}$$

Any convolution $g$ can be approximated by a series of $k$-th order polynomials in the Laplacian, which depend on neighbors within a $k$-hop radius (Hammond et al., 2011). By limiting this approximation to $k = 1$, Kipf & Welling (2017) arrive at an operation that consists of multiplying node features by the normalized adjacency matrix, i.e. averaging each node's neighbor features. Such an operation can be viewed as a graph convolution by a particular smoothing kernel. A Graph Convolutional Network (GCN) is constructed by stacking multiple layers, each with a neighbor averaging step followed by a linear transformation. Many variants of this approach of stacked message-passing layers have since been proposed with different aggregation functions and for different applications (Velickovic et al., 2018; Xu et al., 2018; Hamilton et al., 2017; Schlichtkrull et al., 2018).

Early GNN work focused on tasks with small graphs (thousands of nodes), and it's not straightforward to scale these methods to large-scale graphs. Applying neighbor aggregation by directly multiplying node features by the sparse adjacency matrix at each training step is computationally expensive and does not permit minibatch training. On the other hand, applying a GCN for a minibatch of labeled nodes requires aggregation over a receptive field (neighborhood) of diameter $d$ equal to the GCN depth, which can grow exponentially in $d$. Recent work in scaling GNNs to very large graphs have focused on training the GNN on sampled subsets of neighbors or subgraphs to alleviate the computation and memory cost (Hamilton et al., 2017; Chen et al., 2017; Zou et al., 2019; Zeng et al., 2019; Chiang et al., 2019).

Rossi et al. (2020) proposed a different approach to scaling GCNs, called SIGN: As is shown in Figure 2, by eliding all learned parameters from intermediate layers, the GNN graph aggregation steps can be pre-computed as iterated neighbor feature averages, and model training consists of training an MLP on these neighbor-averaged features. On benchmark tasks on large graphs, they observed that SIGN achieved similar accuracy to state-of-the-art GNNs. The success of SIGN suggests that GNNs are primarily using the graph to "smooth" node features over local neighborhoods rather than learning non-linear feature hierarchies. Similar hypotheses have been argued in several other recent works (Wu et al., 2019; NT & Maehara, 2019; Chen et al., 2019)[1]

Standard GCNs extend naturally to heterogeneous (aka relational) graphs by applying relation-specific learned transformations (Schlichtkrull et al., 2018). There have also been a number of GNN variants specialized to heterogeneous graphs. HetGNN (Zhang et al., 2019a) performs fixed-size random walk on the graph and encodes heterogeneous content with RNNs. Heterogeneous Attention Network (Wang et al., 2019b) generalizes neighborhood of nodes based on semantic patterns (called metapaths) and extends GAT (Velickovic et al., 2018) with a semantic attention mechanism. The Heterogeneous Graph Transformer (Hu et al., 2020b) uses an attention mechanism that conditions on node and edge types, and introduces relative temporal encoding to handle dynamic graphs. These models inherit the scaling limitation of GNN and are expensive to train on large graphs. Therefore, it is of practical importance to generalize the computationally much simpler SIGN model to heterogeneous graphs.

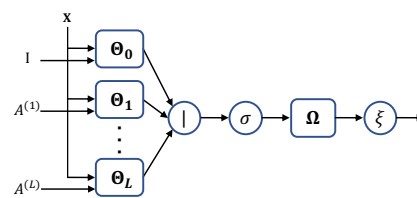

Figure 2: SIGN Model. $A^{(l)}$ is the $l$-th power of adjacency matrix $A$. $\Theta$ and $\Omega$ are transformation parameters in MLP, and $\sigma$ and $\xi$ are non-linear operators.

## 3 NEIGHBOR AVERAGING OVER RELATION SUBGRAPHS FOR HETEROGENEOUS GRAPHS

The challenge with adapting SIGN to heterogeneous graphs is how to incorporate the information about different node and edge types. Relational GNNs like R-GCN naturally handle heterogeneous graphs by learning different transformations for each relation type (Schlichtkrull et al., 2018), but SIGN elides these learned transformations. While one can naively apply SIGN to heterogeneous graphs by ignoring entity and relation types, doing so results in poor task performance (Table 4).

In this section, we propose *Neighbor Averaging over Relation Subgraphs* (NARS). The key idea of NARS is to utilize entity and relation information by repeatedly sampling subsets of relation types and building subgraphs consisting only of edges of these types, which we call relation subgraphs. We then perform neighbor averaging on these relation subgraphs and aggregate features with a learned 1-D convolution. The resulting features are then used to train an MLP, as in SIGN.

We take inspiration from the notion of "metapaths" proposed in Dong et al. (2017) and used by several recent heterogeneous GNN approaches (Wang et al., 2019b). A metapath is a sequence of relation types that describes a semantic relationship among different types of nodes; for example, one metapath in an academic graph might be "venue - paper - author", which could represent "venues of papers with the same author as the target paper node". Information passed and aggregated along a metapath is expected to be semantically meaningful. In previous work like HAN (Wang et al., 2019b), features from different metapaths are later aggregated to capture different neighborhood information for each node.

In prior work, relevant metapaths were manually specified as input (Dong et al., 2017; Wang et al., 2019b), but we hypothesized that the same information could be captured by randomly sampling metapaths. However, sampling individual metapaths doesn't scale well to graphs with many edge

---

[1]SIGN primarily differs from these other proposed methods by concatenating neighbor-aggregated features from different numbers of hops of feature aggregation. This addresses the need to balance the benefits of feature smoothing from large neighborhoods with the risk of "oversmoothing" the features and losing local neighborhood information when GNNs are too deep (Li et al., 2018), allowing the classifier to learn a balance between features from different GNN depths.

types: for a graph with $M$ edge types, it would require $O(M)$ metapaths to even sample each edge type in a single metapath. As a result, one might need to sample a large set of metapaths to obtain good prediction results, and in practice we obtained poor task performance by sampling metapaths.

We observe that metapaths are an instance of a more general class of aggregation procedures: those that aggregate at each GNN layer over a *subset* of relation types rather than a single relation type. We consider a different procedure in that class: sampling a subset of relation types (uniformly from the power set of relation types), and using this subset for all aggregation layers. This procedure amounts to randomly selecting $K$ *relation subgraphs* each specified by a subset of relations, and performing $L$-hop neighbor aggregation across each of these subgraphs. We found that this strategy led to strong task performance (§5), but it's possible that other aggregation strategies could perform even better.

Given a heterogeneous graph $G$ and its edge relation type set $\mathcal{R}$, our proposed method first samples $K$ unique subsets from $\mathcal{R}$. Then for each sampled subset $R_i \subseteq \mathcal{R}$, we generate a relation subgraph $G_i$ from $G$ in which only edges whose type belongs to $R_i$ are kept. We treat $G_i$ as a homogeneous graph, and perform neighbor aggregation to generate $L$-hop neighbor features for each node.

Let $H_{v,0}$ be the input features (of dimension $D$) for node $v$. For each subgraph $G_i$, the $l$-th hop features $H_{v,l}^i$ are computed as

$$H_{v,l}^i = \sum_{u \in N_i(v)} \frac{1}{|N_i(v)|} H_{u,l-1}^i \qquad (2)$$

where $N_i(v)$ is the set of neighbors of node $v$ in $G_i$.

## 3.1 Aggregating SIGN features from sampled subgraphs

For each layer $l$, we let the model adaptively learn which relation-subgraph features to use by aggregating features from different subgraphs $G_i$ with a learnable 1-D convolution. The aggregated $l$-hop features across all subgraphs are calculated as

$$H_{v,l}^{agg} = \sum_{i=1}^{K} a_{i,l} \cdot H_{v,l}^i \qquad (3)$$

where $H^i$ is the neighbor averaging features on subgraph $G_i$ and $a_{i,l}$ is a learned vector of length equal to the feature dimension $D$. In total, $\mathbf{a}$ is a tensor of learned coefficients of size $K \times L \times D$.

We use a 1D convolution to reduce the number of input parameters to the subsequent SIGN classifier, avoiding a multiplicative increase in the number of model parameters by a factor of $K$, as shown in Table 1. Having fewer parameters reduces the cost of computation and memory and is less prone to overfitting.

A classifier is then trained on the aggregated node features to predict task labels, using the MLP architecture described in SIGN (Rossi et al., 2020).

| Model | # parameters |
|---|---|
| SIGN | $D^2 L$ |
| NARS (concat) | $D^2 L K$ |
| NARS (1D conv) | $D^2 L + D L K$ |

Table 1: Number of parameters (equivalently, FLOPs per prediction) in vanilla SIGN and NARS. If features are concatenated in NARS, the number of parameters grows multiplicatively, but this is resolved by using 1D convolution to reduce input dimension to the classifier.

## 3.2 Node types without input features

One consideration for learning on heterogeneous graphs is that input features are not always provided for all node types. Take the OAG academic graph as an example: in prior work, paper nodes were featurized with language embeddings for the paper title using a pretrained XLNet model (Yang et al., 2019). But for other node types like author, field, venue, node features were not provided, so any features must be inferred from the graph.

There are different ways to handle these "featureless" node types, and the best approach might be dependent on the dataset and task. On the tasks we examined, we found it helpful to use relational

| Dataset | Node Types | | | | | Edge Types | | | | |
|---|---|---|---|---|---|---|---|---|---|---|
| | # P | # A | # F | # I | # V | # P-A | # P-F | # P-P | #A-I | # P-V |
| ACM | 4,025 | 17,431 | 73 | – | – | 13,407 | 4,025 | – | – | – |
| OGB-MAG | 736,389 | 1,134,649 | 59,965 | 8,740 | – | 7,145,660 | 7,505,078 | 5,416,271 | 1,043,998 | – |
| OAG (CS) | 5,597,605 | 5,985,759 | 119,537 | 27,433 | 16,931 | 15,571,614 | 47,462,559 | 5,597,606 | 7,190,480 | 31,441,552 |

Table 2: Statistics of three academic graph datasets. The node types are: Paper (P), Author (A), Field (F), Institute (I) and Venue (V)
.

graph embeddings (Bordes et al., 2013) trained on the heterogeneous graph as the initial features for nodes without provided input features. Note that these graph embeddings do not make use of the input features provided for the papers.

We compare the effectiveness of using different types of features for nodes that don't have intrinsic (i.e. content) features in Section 5.4.

# 4 MEMORY FOOTPRINT OPTIMIZATION

NARS allows trainable aggregation of features from sampled subgraphs. However, even though the model is trained in minibatch fashion on GPU, we have to precompute and store all the subgraph-aggregated features in CPU memory. The amount of memory required to store these pre-computed features is $O(NLDK)$, which is $K$ times more than SIGN. So for large heterogeneous graphs with many edge types, there is a tradeoff between sampling more subgraphs in order to capture semantically meaningful relations, and limiting the number of subgraphs to conserve memory during training.

To address this issue, we propose to divide the training into multiple stages. In each stage which lasts several epochs, we train the model with a subset of the sampled subgraphs. Concretely speaking, we approximate Equation (3) with the following equation:

$$H_{v,l}^{(t)} = \sum_{G_i \in S^{(t)} \subseteq S} b_{i,l} \cdot H_{v,l}^i + \alpha H_{v,l}^{(t-1)} \tag{4}$$

In this equation, $S^{(t)}$ is the randomly sampled subset of the $K$ subgraphs used in stage $t$, and $H^{(t)}$ is the approximation of $H^{agg}$ at stage $t$.

Equation (4) is equivalent to setting $a_{i,l}^{(t)} = b_{i,l} + \alpha a_{i,l}^{(t-1)}$ for $G_i \in S^{(t)}$ and $a_{i,l}^{(t+1)} = \alpha a_{i,l}^{(t)}$ for $G_i \notin S^{(t)}$ in Equation (3), so the values of $\mathbf{a}$ can be updated after each stage. The approximated $H^{(t)}$ is used as the input to the classifier during end-to-end training across all stages. The detailed training process is shown in Algorithm 2 in the Appendix.

Our sub-sampling approach reduces memory usage during training to $O(NLD|S^{(t)}|)$, and in practice, we found that even $|S^{(t)}| = 1$ produces decent results and outperforms the accuracy of current SOTA models (see §5.5). One thing worth pointing out is that even though this memory optimization requires regenerating features for $|S^{(t)}|$ subgraphs every a few epochs, the generation can be done on CPU and can completely overlap with GPU training, and hence does not slow down training.

For inference, since all model parameters in Equation (3) are fixed, instead of using our proposed memory optimization method, we simply generate features for each sampled subgraph and compute the aggregation in place. The memory overhead for inference is therefore also only $O(NDL)$.

# 5 EVALUATION

In this section, we evaluate NARS on several popular heterogeneous graph datasets and compare with state-of-the-art models. We also investigate the effect of different ways to featurize nodes without input features, and how the memory optimization described in §4 affects prediction accuracy.

## 5.1 EXPERIMENTAL SETUP

**Datasets & Tasks**  We evaluate our model using node prediction on three popular academic graph datasets: ACM (Wang et al., 2019b), OGB-MAG (Hu et al., 2020a), and OAG (Sinha et al., 2015;

| Dataset | Hyper-parameters | | | | # Model Parameters | | | | |
|---|---|---|---|---|---|---|---|---|---|
| | # hidden | # layers (L) | # subgraphs (K) | TransE size | NARS (Ours) | SIGN | R-GCN | HAN | HGT |
| ACM | 64 | 2 | 2 | 128 | 0.40M | 0.39M | 0.14M | 0.25M | 0.26M |
| OGB-MAG | 512 | 5 | 8 | 256 | 4.13M | 4.12M | 9.18M | – | 26.88M |
| OAG-Venue | 256 | 3 | 8 | 400 | 2.24M | 2.21M | 40.60M | – | 8.26M |
| OAG-L1-Field | 256 | 3 | 8 | 400 | 1.41M | 1.38M | 11.64M | – | 7.43M |

Table 3: Training hyper-parameters & number of parameters for each model. For hyper-parameters # hidden and # layers, we adopt values from HAN for ACM, and values from HGT for OGB-MAG and OAG. # subgraphs sampled for NARS is picked to be small while producing good and stable results. TransE size is selected based on the size of the graph.

| Dataset | ACM | OGB-MAG | OAG-Venue | | OAG-L1-Field | |
|---|---|---|---|---|---|---|
| Metric | Accuracy | Accuracy | NDCG | MRR | NDCG | MRR |
| R-GCN | **.930±.002** | .500±.001 | .481±.004 | .302±.005 | .852±.002 | .843±.002 |
| HAN | .922±.002 | – | – | – | – | – |
| HGT[3] | .919±.003 | .498±.001 | .498±.014 | .322±.014 | **.868±.002** | .849±.003 |
| SIGN | .919±.001 | .481±.001 | .506±.001 | .327±.001 | .839±.001 | .826±.003 |
| NARS (Ours) | **.931±.004** | **.521±.004** | **.520±.003** | **.342±.003** | **.868±.001** | **.857±.003** |

Table 4: Performance of NARS vs. baseline models on different datasets and tasks. All numbers are average and standard deviation over 5 runs. Bold numbers indicate best model(s).

Tang et al., 2008; Zhang et al., 2019b). The tasks involve predicting a paper's category, its publishing venue, or its research field on these three datasets. We summarize the statistics of the datasets in Table 2 and put the details of each dataset in Appendix (A.3).

**Baselines & Metrics** We compare NARS with four baseline models: R-GCN (Schlichtkrull et al., 2018), HAN (Wang et al., 2019b), HGT (Hu et al., 2020b), and SIGN (Rossi et al., 2020). The first three models are designed for heterogeneous scenarios and can be naturally applied on all datasets. The vanilla SIGN model, however, is for the homogeneous graph setting. Therefore, we ignore the node types and edge types when training SIGN. We use the best implementation we can find for each baseline model: for HGT, we use the authors' original implementation. For SIGN and HAN, we use the implementation by Deep Graph Library (Wang et al., 2019a). For R-GCN, we use the implementation by PyTorch Geometric (Fey & Lenssen, 2019).

We report test accuracy (micro-F1) for the ACM and OGB-MAG datasets and NDCG and MRR for the OAG dataset, using the model from the epoch that performs best on the validation set. For each experiment, we run five replicates and report the average and standard deviation across replicates.

Training GNN baselines on the OGB-MAG and OAG datasets is intractable due to memory usage, unless sampling is used. We adopt neighbor sampling (Hamilton et al., 2017) for R-GCN and HGT's custom sampling method, but were unable to evaluate HAN on these datasets, since it is unclear how to train this model with sampling while following metapath constraints.

**Training settings** We train NARS with learning rate 0.001, dropout rate 0.5 and the Adam optimizer. For nodes that don't have input features, we use pre-trained TransE embeddings (Bordes et al., 2013) as input. During the preprocessing phase, we sample $K$ subgraphs and pre-compute the aggregated features of $L$ hop neighbors for each node. Table 3 shows the training hyper-parameters for our model, as well as a comparison of model sizes (in terms of the number of parameters) across ours and various baseline models. Though we try to use the same hyper-parameters for all models, we note that our extension to SIGN results in a simple model and in most cases has significantly fewer training parameters.

Detailed implementation and configuration can be found in our open-sourced GitHub repository[2].

## 5.2 RESULTS

---

[2]To be released

[3]We improved the reported performance of HGT by 5% by sampling 10 times for target nodes and using the average of predictions, to reduce sampling variance.

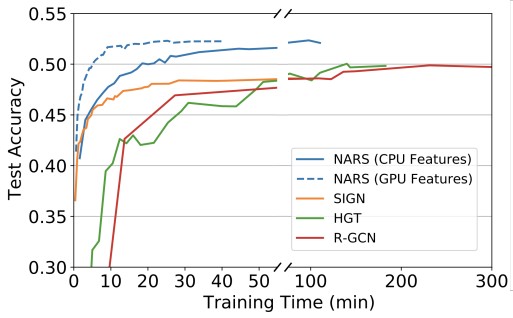 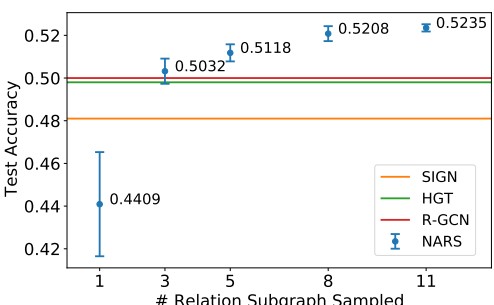

Figure 3: Training speed of NARS (sampling 8 subgraphs) and baseline models on the OGB-MAG dataset. NARS leads to higher final accuracy in less training time. NARS training is much faster when features can be stored on GPU (dashed line).

Figure 4: Accuracy of sampling different numbers of relation subgraphs on the OGB-MAG dataset. Sampling more relation subgraphs improves test accuracy while reducing variance across replicates.

Table 4 summarizes the performance of NARS and baseline models on different datasets and tasks. NARS outperforms all baseline methods on all datasets and node prediction tasks that we evaluated. In particular, NARS improves test performance by up to 4% compared to the current state-of-the-art model (HGT) on large datasets like OAG and MAG. Compared to naively applying SIGN by treating the graph as homogeneous, NARS improves performance by up to 8%. It is quite surprising that NARS exceeds the performance of existing state-of-the-art GNN methods since NARS is computationally much simpler than the baselines; like SIGN, it does not learn any feature hierarchies over graph neighborhoods, but merely learns a classifier on graph-smoothed features.

Figure 3 compares the training speed of NARS and public implementations of baseline methods on the OGB-MAG dataset. We did not include the time for pre-training TransE embeddings (40 min) since all models benefit from using them (§5.4). In Figure 3, the solid blue line labelled NARS (CPU features) refers to an implementation that stores all pre-computed features on the CPU and transfers them to the GPU on demand during each mini-batch training. This NARS implementation not only achieves superior accuracy, but also is substantially faster to train than competing approaches. We can further improve NARS training speed by storing pre-computed features on the GPU if space permits, as shown by the dotted blue line labeled NARS (GPU features).

## 5.3 Effect of number of subgraphs sampled

We have shown in Table 4 that sampling relation subgraphs produces better results than treating the graph as homogeneous. Now we use the OGB-MAG dataset to perform an ablation study on how the number of sampled subgraphs affects the performance.

OGB-MAG dataset has 4 edge relation types, hence the edge relation set $R$ has in total 15 non-empty subsets. Among the 15 subsets, only 11 are valid (others don't touch paper nodes for prediction or are disconnected). In Figure 4, we randomly sample subsets of size 1, 3, 5, 8 from $R$ which contains the 11 valid subgraphs. For each point in the figure, we randomly sample 3 different subsets and report the average and standard deviation.

As shown in Figure 4, the average accuracy improves while the variance decreases as the number of sampled subgraphs increases, which is expected since sampling more reduces sampling variance and increase the chance to cover relation subgraphs that are more semantically meaningful. On the OGB-MAG dataset, sampling $\geq 5$ subgraphs always outperforms current state-of-the-art models like HGT (green line) and R-GCN (red line).

## 5.4 Effect of different strategies to handle featureless nodes

In this section, we compare how different ways of featurizing nodes that don't have input features affect the performance of NARS on OGB-MAG dataset. In the dataset, only paper nodes have input language features generated using word2vec, and all other node types are not associated with any input features.

|  | Padding Zeros | Average Neighbors | Metapath2vec Embs | TransE Embs |
|---|---|---|---|---|
| NARS | .4462±.0022 | .4427±.0018 | .5187±.0011 | .5214±.0016 |
| SIGN | .4007±.0010 | .4028±.0021 | .4685±.0011 | .4810±.0009 |
| HGT | .4597±.0027 | .4891±.0021 | .4962±.0026 | .4982±.0013 |
| R-GCN | .4811±.0028 | .4707±.0033 | .5013±.0019 | .5001±.0005 |

Table 5: Comparison of different ways to featurize nodes with no input features on the OGB-MAG dataset. All models achieve their best performance with pre-trained TransE graph embedding features. Featurization is especially important for neighbor-averaging approaches (SIGN and NARS).

We tried four different ways to generate features for the featureless nodes: 1) padding zero; 2) taking the average of features from neighboring papers nodes; 3) using pre-trained Metapath2vec embedding; 4) using pre-trained TransE relational graph embeddings with L2 norm. We followed the same hyper-parameters listed in Table 3.

As shown in Table 5, unsupervised graph embedding methods greatly improve model accuracy, especially for SIGN and NARS. We use TransE embeddings in the rest of this section because it achieves the best accuracy, and because metapath2vec embeddings require manually specifying metapaths, which we seek to avoid.

## 5.5 EFFECT OF TRAINING WITH SUBSET OF SAMPLED SUBGRAPHS

In section 4, we proposed to train with a subsampled set from all $K$ sampled relation subgraphs to reduce CPU memory usage. In Figure 5, we vary the number of relation subgraphs in the randomly sampled subset in each stage on OGB-MAG dataset to see how this approach affects accuracy. The blue dashed line at the top of the figure is the accuracy for training with all 11 valid relation subgraphs. The blue point with error bar represents the average and standard deviation for sampling a certain number of subgraphs from the 11 valid relation subgraphs in each stage. Performance improves when more subgraphs are sampled in each stage, but sampling a single subgraph in each stage leads to good performance with small variance, outperforming existing models.

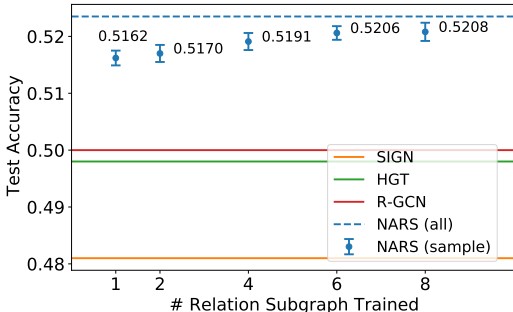

Figure 5: Training with different numbers of subgraphs in each stage (§4) on OGB-MAG dataset, to improve memory efficiency. Good test accuracy is achieved with even a single subgraph sampled per training stage.

## 6 CONCLUSION

Simplified GNNs like SIGN that do not require learned aggregation are a promising new class of models for graph learning due to their simplicity, scalability, and interpretability. We present Neighbor Averaging over Relation Subgraphs, a novel GNN architecture for heterogeneous graphs that learns a classifier based on a combination of neighbor-averaged features for random subgraphs specified by a subset of relation types. NARS beats state-of-the-art task performance on several benchmarks despite its simpler and more scalable approach.

This work provides further evidence that current GNNs might not learn meaningful feature hierarchies on benchmark datasets, but are primarily functioning by graph feature smoothing. Future advances in GNN modeling may realize the benefits of "deep" learning for these tasks, or it may be that this modeling is not necessary for many important graph learning tasks.

One remaining limitation of NARS is that it doesn't explicitly handle heterogeneous feature types, averaging together features of different types (e.g. language features vs. graph features). While this performs adequately on benchmark datasets, it is an unsatisfying approach for industry graph datasets with potentially dozens or hundreds of distinct entity types, each with their own distinct features. In these situations, existing GNNs (e.g. R-GCN) may be more appropriate, and neighbor averaging approaches suitable for this situation are an area for future work.

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

# A APPENDIX

## A.1 ALGORITHM TO TRAIN NARS

Algorithm 1 demonstrates the detailed training algorithm for Neighbor Averaging over Relation Subgraphs.

---

**Algorithm 1:** Neighbor Averaging over Relation Subgraphs

---

**Input:** $G$ a heterogeneous graph
**Input:** $\mathcal{R}$ the set of relations in $G$
**Input:** $K$ number of relation subgraphs to sample
**Input:** $L$ number of layers in MLP model
**Input:** $H_{input}$ node input features

```
/* Generate multi-hop neighborhood-averaged features        */
```
**def** *GenNeighborFeature(G)*:
```
    // Build adjacency matrix for G regardless of edge types
```
    $A \leftarrow BuildAdjacencyMatrix(G)$;
```
    // Divide nonzeros in each row by node in-degree
```
    $W \leftarrow RowNormalize(A)$;
    $H_0 \leftarrow H_{input}$;
    **for** $l \leftarrow 1$ **to** $L$ **do**
        $H_l \leftarrow WH_{l-1}$;
    **return** $H_0, H_1, \ldots, H_L$;

```
/* Sample K unique relation subgraphs from G and R          */
```
**def** *SampleRelationSubgraph(G, $\mathcal{R}$, K)*:
    $P \leftarrow GetPowerSet(\mathcal{R})$;
    $S \leftarrow SampleWithoutReplacement(P, K)$;
    $E \leftarrow GetEdgeSet(G)$;
    **for** $R_i \in S$ **do**
        $E_i \leftarrow \emptyset$;
        **for** $e \in E$ **do**
            **if** $e.type \in R_i$ **then**
                $E_i \leftarrow E_i + \{e\}$;
        $G_i \leftarrow BuildSubgraphFromEdgeSet(E_i)$;
    **return** $G_0, G_1, \ldots, G_{K-1}$;

```
/* Prepreocssing                                            */
```
$S \leftarrow SampleRelationSubgraph(G, \mathcal{R}, K)$;
**for** $G_i \in S$ **do**
    $H_0^i, H_1^i, \ldots, H_L^i \leftarrow GenNeighborFeature(G_i)$
```
/* Training                                                 */
// Any MLP classfier, SIGN model for example
```
$model \leftarrow MLPModel()$;
**for** $epoch \leftarrow 1$ **to** *MAX_EPOCH* **do**
```
    // Compute Equation 3
```
    **for** $l \leftarrow 0$ **to** $L$ **do**
        $H_l \leftarrow \sum_{i=0}^{K} a_l^i H_l^i$;
    $loss \leftarrow model.forward(\{H_l\})$;
    $loss.backward()$;
    $gradient\_update(\{a_l^i\}, model)$;

---

## A.2 Algorithm to train with reduced memory usage

Algorithm 2 demonstrates details about how to further sub-sample $p$ subgraphs in each stage from the set of relation subgraph $S$ and use Equation (4) to train the model. In Algorithm 2, we omit the layer number $l$ to make it more concise.

## A.3 Datasets

**ACM**    We use the ACM dataset from Heterogeneous Graph Attention Network (Wang et al., 2019b), which is an academic graph extracted from papers published in ACM conferences (KDD, SIGMOD, SIGCOMM, MobiCOMM, and VLDB). The HAN author-provided version removed the field and author node types. Therefore, we used the version re-constructed by DGL (Wang et al., 2019a). In this dataset, only paper nodes have Bag-of-Words features. Papers are divided into three classes and the task is to predict the class for each paper.

**OGB-MAG**    Open Graph Benchmark (Hu et al., 2020a) is an effort to build a standard, large and challenging benchmark for graph learning, which contains a large collection of datasets that cover important tasks on graphs and a wide range of domains. Leaderboards are set up for each dataset and performance of state-of-the-art models is listed with open-sourced implementation to reproduce results. We evaluate our approach on the MAG benchmark from OGB node prediction category, which is a heterogeneous network extracted from Microsoft Academic Graph (MAG). The papers are published on 349 different venues and they come with Word2Vec features. The task here is to predict the publishing venue for each paper.

**OAG**    Open Academic Graph (Sinha et al., 2015; Tang et al., 2008; Zhang et al., 2019b) is the largest public academic graph with more than 178 million nodes and 2 billion edges. We use the pre-processed CS domain component[4] provided by the authors of Heterogeneous Graph Transformer (HGT) (Hu et al., 2020b) in order to have a fair comparison with HGT. Field nodes are further divided into 6 levels (L0 to L5) and as a result, the graph comes with 15 edge types, adding rich edge relation information to the graph. Each paper is featurized with a language embedding generated by pre-trained XLNet on its title. Following HGT, we evaluate two tasks on paper nodes: predicting publishing venues and predicting L1 field (multi-label). One potential issue with OAG dataset is indirect information leakage since target nodes have edges connecting to ground truth label nodes in OAG graph. To address this issue, for each task, we remove all edges between paper nodes and corresponding label nodes we are to predict. For example, if the task is predicting venues of papers, we remove all edges between paper nodes and venue nodes.

---

[4]HGT authors only shared the CS component. Also they performed a filter step to make the graph denser. After the filtering, there are 544244 papers, 45717 fields, 510189 authors, 9079 institutes, and 6933 venues left.

---

**Algorithm 2:** Train with subset of sampled subgraph

---

**Input:** $S$ a set of $K$ sampled subgraphs
**Input:** $T$ number of epochs in each stage
**Input:** $p$ number of subgraphs used for training in each stage
$model \leftarrow MLPModel()$;
$Uniform\_Random\_Init(\{a_k\})$;
// Initialize history of aggregated features using Equation 3
$H^{(0)} \leftarrow 0$;
**for** $G_i \in S$ **do**
    $H^i \leftarrow GenNeighborFeature(G_i)$;
    $H^{(0)} \leftarrow H^{(0)} + a_i H^i$;
// Sample a subset $S^{(0)}$ from $S$
$S^{(0)} \leftarrow SampleSubset(S, p)$;
**for** $G_i \in S^{(0)}$ **do**
    // Generate neighbor feature for each sampled $G_i \in S^{(0)}$
    $H^i \leftarrow GenNeighborFeature(G_i)$;
    // Initialize $b_i$ to 0
    $b_i \leftarrow 0$;
// Initialize $\alpha$ to 1
$\alpha \leftarrow 1$;
// Initialize stage $t$ to 1
$t \leftarrow 1$;
**for** $epoch \leftarrow 1$ **to** $MAX\_EPOCH$ **do**
    // Compute approximated aggregation following Equation 4
    $H^{(t)} \leftarrow \alpha H^{(t-1)}$;
    **for** $G_i \in S^{(t)}$ **do**
        $H^{(t)} \leftarrow H^{(t)} + b_i H^i$;
    $loss \leftarrow model(H^{(t)})$;
    $loss.backward()$;
    $gradient\_update(model, \{b_i\}, \alpha)$;
    **if** $epoch \% T == 0$ **then**
        // Update history of aggregation
        $H^{(t)} \leftarrow \alpha H^{(t-1)}$;
        **for** $G_i \in S^{(t)}$ **do**
            $H^{(t)} \leftarrow H^{(t)} + b_i H^i$;
        // Update $a_i$
        **for** $G_i \in S^{(t)}$ **do**
            $a_i \leftarrow b_i + \alpha a_i$;
        **for** $G_i \notin S^{(t)}$ **do**
            $a_i \leftarrow \alpha a_i$;
        // Increment stage $t$
        $t \leftarrow t + 1$;
        // Re-sample $S^{(t)}$ from $S$
        $S^{(t)} \leftarrow SampleSubset(S, p)$;
        **for** $G_i \in S^{(t)}$ **do**
            // Generate neighbor feature for newly sampled $G_i \in S^{(t)}$
            $H^i \leftarrow GenNeighborFeature(G_i)$;
            // Re-initialize $b_i$ to 0
            $b_i \leftarrow 0$;
        // Re-initialize $\alpha$ to 1
        $\alpha \leftarrow 1$;

---

