# OpenReview forum: "Scalable Graph Neural Networks for Heterogeneous Graphs"
_ICLR.cc/2021/Conference — Reject_

### Official Review · AnonReviewer2 · 2020-10-28
**Broadening the scope of SIGN to multi-relational graphs**

**Rating:** 6
**Confidence:** 4

**Review:**

The authors propose a method to broaden the scope of SIGN, a technique recently introduced for single-relational graphs. The method allows SIGN to also be applied to multi-relational graphs (often called heterogeneous or knowledge graphs in different communities). In SIGN, various powers of the Laplacian are precomputed. For each power, the features of the nodes of a node’s neighborhood are averaged and (e.g., with an MLP) projected into a node vector representation. This is then used to classify the node.

One way to apply SIGN to multi-relational graphs would be to conflate all relation types into a single relation type. The authors show that this doesn’t work well.

The authors then propose the following: take the set of all relation types R and sample a subset R’ from it. Construct the subgraph induced by R’ by only keeping edges of relation types in R’. Then treat all of these relation types as one (turn into a single-relational subgraph) and apply (essentially) SIGN to this subgraph. Do this several times, that is, sample subsets of R several times. Finally aggregate the representations coming from each of the application of SIGN to the sampled subgraphs.

The authors then also discuss what to do if the nodes do not have attributes (features) and ways to improve the memory efficiency of the approach.

The authors then proceed to empirically evaluate their method. They compare to existing approaches (such as R-GCNs) and show that NARS (the name of their approach) is competitive with these existing methods. Often it is significantly better. They also perform several types of ablation studies, analyzing the impact of choices such as the number of subgraphs sampled. Generally, I would say that the experiments are well done and look at various different important questions.

In summary, I don’t have much to criticise. The only shortcoming is that the method is more or less an adaptation of SIGN to the multi-relational setting. I want to be careful to lament a “lack of novelty” here, because it is still a good contribution to broaden the scope of an existing method to a larger class of problems. However, one thing I’m missing is a more thorough analysis of the sampling strategies one could use. What is analyzed is the number of subgraphs sampled. But this is based on one way of sampling subgraphs (based on first sampling relation types and inducing the graph based on this). What I’m missing are: sampling subgraphs from the Gaifman graph; and sampling multi-relational graphs and applying R-GCN on each of those and aggregating. The latter is probably less efficient, but I would conjecture also much better than running one R-GCN. Again, see related work below. It has been shown before that sampling multi-relational subgraphs can add robustness of the model to overfitting. Overall, I would tend towards accepting the paper but I would really like to see more analysis on different sampling strategies.

I would encourage the authors to consider as related work a paper from 2017 which is one of the first to propose neighborhood sampling for GNNs (much earlier than most papers you mention in this context) and which is one of the first ones who simply average (learned) features but with an unsupervised objective:

Garcia-Duran and Niepert, Learning Graph Representations with Embedding Propagation, NeurIPS 2017

---

> ### Author Response · Authors · 2020-11-19
> **Response to Reviewer #2**
>
> We thank the reviewer for the positive and constructive feedback.
>
> We agree with the reviewer that it would be interesting to explore different approaches to subgraph sampling, although these alternative strategies may sacrifice some of the simplicity of NARS.
> The proposed experiment with R-GCN on subgraphs is interesting and we will add it to the camera-ready. From our perspective, avoiding the need to do message passing during training is a primary benefit of NARS, which would not be the case for R-GCN; however, it would be informative to see if the benefits of subgraph sampling still apply when relational operators can be learned. Regarding Gaifman graphs, are you proposing first generating the Gaifman graph for the multi-relational graph and then sampling some subgraphs of that (homogeneous) graph? Which ones?
>
> Re related work, thanks for pointing us at this earlier instance of negative sampling, we weren’t aware of it. We’ll mention it in the prior work on sampling methods for large graphs.

---

> > ### Comment · AnonReviewer2 · 2020-11-19
> > **Re: Response to Reviewer #2**
> >
> > Dear authors,
> >
> > Thank you for responding to my review. I have also read the other reviews. First, I want to encourage you to continue this work irrespective of the outcome here. I disagree with some other reviewers who've simply stated that the paper should be rejected because of "a lack of novelty." What matters is that a paper provides novel understanding of a problem and not necessarily a novel method. To strengthen your paper, however, I would really encourage you to explore more than one way to induce (sample) subgraphs. You proposed one such way. A stronger paper would set out to explore and evaluate several ways to sample subgraphs.  A really interesting (and to me exciting) question would also be to relate your sampling method to the types of (implicit) rules your method can learn. It is known that rule-based methods work really well on multi-relational graphs. Rules usually involve a small number of relation types (at most 2 for Horn rules with 2 elements in the body) and your method might make it easier for the model to implicitly learn such rules from the sampled subgraphs. This might be an interesting "qualitative" study: looking through attribution methods (e.g. integrated gradients or such) at the types of relation type combinations the model is leveraging and to what rules this might correspond.
> >
> > Overall, I will not lower my score but I also don't want to increase it as for this I would want to see a more thorough evaluation of more sampling strategies.

---

> > > ### Author Response · Authors · 2020-11-24
> > > **Re: Re: Response to Reviewer #2**
> > >
> > > We appreciate the encouragement and constructive feedback. We are working on comparing alternative metagraph sampling approaches and will add these results to the next version of the paper. Regarding some of the more complex sampling strategies you propose, we don't think the existing benchmarks have complex enough multi-relational structure to provide meaningful results, but we'd be curious to look at them for node classification on knowledge graphs, for example.

---

### Official Review · AnonReviewer1 · 2020-10-28
**Interesting Problem, but the novelty of the technical depth is marginal**

**Rating:** 3
**Confidence:** 5

**Review:**

This paper studied heterogeneous graph embedding with graph neural networks. The authors proposed an approach which samples multiple different subgraphs containing a subset of relations and then aggregate the node representations from different subgraphs with attentions. Experimental results on some big data sets prove the effectiveness and efficiency of the proposed approach.

Overall, heterogenous graph embedding with graph neural network is a very important and interesting problem with a variety of applications. However, the novelty of the proposed approach seems to be quite marginal, and I am not quite convinced by the proposed approach. Why are we able to get better results by first sampling a subset of relations to get the subgraph and then aggregating the node representations from different subgraphs? It seems weird to be as a randomly selected subset of relations do not convey any specific semantic meanings as traditional metapath based approaches. I am also surprised by the very small standard deviation in Table 4, which is not consistent with results reported in existing literature on the tasks of node classification.

---

> ### Author Response · Authors · 2020-11-19
> **Response to Reviewer #1**
>
> We thank the reviewer for the response.
>
> Regarding novelty, we do not believe NARS is an obvious extension of prior work, and indeed the fact that the reviewers are surprised (as we were) that this model achieves such strong performance suggests that it’s an interesting and non-obvious result. We agree that NARS is a *simpler* approach than competing methods like HGT and HAN, but we see that as a benefit of this approach, considering that it achieves superior performance to these methods while being simpler, faster and more scalable.
>
> The reviewer also asks why NARS works when the randomly-sampled relation subgraphs are not semantically meaningful. Our goal with NARS was to allow the model to learn which random “metapaths” are semantically meaningful (through the learned attentions) rather than providing them as an input as in prior approaches, although in principle NARS certainly allows the subgraphs/metapaths to be provided by the user. We were also a bit surprised that we were able to achieve SotA accuray by randomly selecting subgraphs and learning an attention over them, thus removing the need for hand-tuning metapaths.
>
> For the small standard deviation the reviewer mentioned, we are not sure which papers the reviewer is referring to. The standard deviation we got has similar scale to our baselines except HGT, which performs sampling during testing. Also, the scale of std of our results is consistent with those on node prediction leaderboard of OGB benchmark: https://ogb.stanford.edu/docs/leader_nodeprop/

---

### Official Review · AnonReviewer3 · 2020-10-29
**Review #3**

**Rating:** 5
**Confidence:** 5

**Review:**

This paper extends from SIGN (https://arxiv.org/abs/2004.11198) model to heterogeneous graphs.

The SIGN model argues that simply applying MLP on graph-smoothed node features (concatenating k-th hop neighbor features, k $\in$ [1-L]) can achieve similar results compared with learnable aggregation applied in GNNs. To extend to the heterogeneous graph, this paper proposes to sample relation graphs, by:
(1) sample several subsets $R_i$ of relations;
(2) sample relation subgraphs whose edges belong to $R_i$;
(3) treat each subgraph as homogeneous graphs and perform neighbor aggregation (simply average).
(4) Apply MLP on each node for node classification.

I have several questions about the proposed approaches:

(1) The difficulty of heterogeneous graphs is that each node might have different types of features. For example, in a social network, nodes can be associated with image, text, or some discrete profiles. Thus, the neighborhood smoothing only works when the input features are both 1) already very informative, and don't need too much transformation; 2) features from different node types are projected to the same space. Therefore, I'm afraid the authors' proposed aggregation might not generalize to more complicated heterogeneous graphs. (It seems that during experiments, the authors utilize TransE to pre-train embeddings for all the nodes, so that they are naturally within the same space, making the problem simpler. One evidence is that when using other feature initialization strategies, such as simple average, the performance of this model drops significantly)

(2) Also, it's confusing to me why can we fuse all the subgraphs with different subgraph schema. Intuitively, with different relation set, the semantic of the relation subgraph should be very different, but the authors seem to treat them equally. It would be better if the authors can provide some analysis on this, for example, for a given node, what is the variance of final node embeddings calculated with subgraphs of different relation sets.

(3) How to get the inference results for large graphs? It seems that the proposed method should get a different predictions for each node with a different relation set. So which set the authors to use? Complete set or average over random sampling? (If it's random, the reported variance, which is close to 0, seems to be very strange).

Also, though the authors show superior experimental results, I have several concerns about experiment settings:

(1) About feature initialization. From Table 5, we can see that the proposed NARS method highly relies on the TransE
 embedding initialization. When using a standard feature initialization method (such as average neighbors), the results are much lower than HGT and R-GCN. However, the authors didn't provide implementation details about how to train such TransE embedding (normally it's weird to use TransE for heterogeneous graph, as the node number can be much larger than the knowledge graph and we don't have that much relation type. For example, two papers published by the same author and on the same venue might have exactly the same TransE embedding, if we don't consider text input. So it's confusing to me why the results with TransE embedding is better). The authors should better elaborate on this part or release the code for clarity.

(2) About baseline results. Since the utilization of TransE embedding, the experimental settings of the baseline are different from the original papers. But there's still some confusing part. For example, the HGT model's implementation on OGB-MAG uses neighbor average strategy, and the accuracy result is 0.5, while the result reported in table 5 is 0.489. Also, the model parameter is not matched with the reported number.

(3) About inference time. As discussed above, I'm not sure how the proposed method can efficiently get accurate inferences for all the nodes in the test set. If the authors want to claim their method is more scalable, it would be better to include the inference time comparison.


Overall, I think the simplified procedure (direct neighbor average) over heterogeneous graph limits the usage of this model, and there's also some unclear part in experimental settings.

---

> ### Author Response · Authors · 2020-11-19
> **Response to Reviewer #3**
>
> We thank the reviewer for the detailed feedback.
>
> The reviewer’s summary description of the method is correct but misses a crucial step which may be responsible for some misunderstandings:
> After step (3), we *aggregate* the neighbor-aggregated features using a learned 1D convolution into a single feature set, which is then fed to the MLP for node classification. You can think of this as learned attention weights over the different subgraph features.
>
> Part A: for reviewer’s questions regarding the proposed approaches:
> 1. “Generalization to more complex graphs with multiple feature types:”
> We agree with the reviewer that it’s not clear how well this technique will generalize to graphs that are more complex than any published benchmark datasets, especially when different entities have different types of features. One promising sign is that NARS is successful on these benchmark academic graph tasks when a different type of feature is used for paper nodes (text features) and non-paper nodes (graph embedding features). Alternatively, multiple types of features can be treated as a single feature type that is the concatenation of all feature types (padding zeros when a node is missing features of a type), but of course that doesn’t scale to large numbers of distinct features types.
> Regarding initialization with TransE, we only do this for nodes that do not come with input features, which is consistent with what prior work has done (although they used metapath2vec rather than TransE embeddings). The reviewer is correct that NARS is more sensitive to missing input features than competing methods, presumably because it can’t learn structural information. But TransE features are cheap to generate relative to HGNNs, and you can generate them once in advance and use them for multiple tasks, so we feel that this is a good tradeoff. We view this comparison of different input embedding types as a contribution of our work, since it represents an effective way to improve (especially) neighbor-averaging approaches without sacrificing scalability.
> 2. “How to aggregate subgraphs?”: The aggregation of features from different relation subgraph schema is done by a learned convolution, and the idea here is to give the model the power to choose which relation subgraphs are more meaningful. This is similar to the semantic-level attention mechanism of Heterogeneous Graph Attention Network where features generated by different semantic meta-paths are aggregated by an attention weight. Even though different features have different semantic meanings, they can be aggregated together as long as they are in the same embedding space.
> 3. “How to do inference?”: At inference time, we compute neighbor-averaged features for all of the subgraphs sampled during training and aggregate them with the learned 1D convolution. Figure 4 illustrates how sensitive the performance is to the particular set of subgraphs sampled; as you increase the number of subgraphs it becomes less sensitive.
>
> Part B: for reviewer’s questions regarding the experiment settings
> 1. “Feature initialization”: As we mention in response A(1), we only use graph embeddings for nodes without input features, which is consistent with prior work (e.g. HGT). We use the existing DGL-KE package (https://github.com/awslabs/dgl-ke) to train TransE with no modifications. A comparison of different input embeddings is provided in Table 5; we find that metapath2vec embeddings work okay as well. We have released code so that all of these steps can be replicated.
> 2. “Baselines reported performance:” Thanks for pointing this out. The reason for the inconsistency is that we used 5 rather than 4 layers for HGT on the MAG datasets, which leads to overfitting when using neighbor-averaged features. This overfitting mostly goes away once we add embedding features, but we will update Tables 4 and 5 to reflect the improved HGT performance.
> 3. “Inference”: Thanks for mentioning this omission, we will add inference compute time comparisons to the paper. For the OGB-MAG dataset, NARS takes about 5 seconds to predict labels for all nodes, compared to ~30 minutes for the HGT reference implementation. In general, however, properly implemented GNN inference is typically very cheap compared to training because message passing can be performed a single time to compute labels for all nodes; NARS’ training speedup comes from its ability to do this at training time as well.

---

> > ### Comment · AnonReviewer3 · 2020-11-24
> > **Response**
> >
> > Thanks for the detailed response.
> >
> > I do appreciate the authors' results with a very simple model design. But at the current stage, I'm not that confident whether the improved performance is achieved with the informative TransE input, as the authors admit overfitting might happen for the other baseline models. And also, I'm more willing to see the evaluation on datasets with different types of input features, which would definitely make the paper more convincing and useful for heterogeneous graph mining.
> >
> >
> > The faster inference speed and the simple model design is a good part of the paper. Due to the authors' clarification, I can raise the score to 5. Looking forward to the complete experimental results (e.g., inference time comparison, consistent evaluation on OGBN-MAG)

---

### Official Review · AnonReviewer4 · 2020-10-30
**This paper aims to propose a new GNN for heterogeneous graphs, which is scalable to large-scale graphs. The proposed idea is to leverage an existing model called SIGN, which simplifies GCN by dropping the non-linear transformation from intermediate layers, and extend it to heterogeneous graphs. The results on several benchmark datasets show the proposed approach is better and faster than baselines.**

**Rating:** 5
**Confidence:** 5

**Review:**

This paper aims to propose a new GNN for heterogeneous graphs, which is scalable to large-scale graphs. The proposed idea is to leverage an existing model called SIGN, which simplifies GCN by dropping the non-linear transformation from intermediate layers, and extend it to heterogeneous graphs. The results on several benchmark datasets show the proposed approach is better and faster than baselines.

Although the proposed idea seems interesting, there are several concerns about the paper.

1.	The description of the methodology is very vague. For example, Fig. 1 is presented without detailed explanation. It is unclear how the features computed by different subgraphs can be aggregated, especially consider nodes only appear in a subset of those subgraphs. The formula in Eq. (2) and (3) do not help due to their simplicity.
2.	The novelty of the paper is also limited, consider it is extending an existing algorithm SIGN to heterogeneous version.
3.	I am not fully convinced the simplified GNN works better than some other GNNs designed for heterogeneous networks, such as HAN and HGT, which contains attention scheme and carefully models the message type in the GNN framework. According to other simplified GNN paper, their performance is just comparable to their counterpart but not better, and usually the results are worse than the attention-based one.
4.	It seems the numbers in Table 4 is higher than the ones in the original paper, such as HGT.

---

> ### Author Response · Authors · 2020-11-19
> **Response to Reviewer #4**
>
> We thank the reviewer for the feedback.
> 1. “How to aggregate features”: Thanks for pointing out the lack of clarity. The features are aggregated by a 1D convolution (Sec 3.1); i.e. each output feature is a linear combination of the features from the different subgraphs with learned weights. Every node is in every subgraph (subgraphs are subsets of edges); if a node has no neighbors then its features in that subgraph will be a vector of 0s.
> 2. “Novelty”: NARS is a new SotA method for HG node classification that is also simpler and more scalable than existing approaches. This task is important enough to justify substantial prior research published at similar venues (e.g. HGT, HAN). We don’t think it was obvious a priori how neighbor-averaging approaches like SIGN could be extended to HGs, or that an approach like NARS would work (in fact, several reviewers are surprised that this approach should work). We see value to the ICLR community for this work since it produces SotA performance on HG classification, and is simpler and more scalable than existing approaches.
> 3. “Performance vs. HAN and HGT”: We agree with the reviewer; we were also surprised that NARS achieves superior performance to existing methods, which are much more complex. As a result, we were very careful with our baseline comparisons; in fact, we were able to improve the performance of some of the baselines relative to those reported (as the reviewer points out). We’re not sure what would convince the reviewer besides our reported results, but to that end we have open sourced the code for these experiments and are submitting the results to the OGB leaderboard so they can be verified by the community.
> 4. “Higher number than reported in the original paper”: We ran the official implementation of HGT on the same datasets used in the original paper. Since we were surprised by the performance of our model, we spent some time improving the baselines from the reported results. Specifically, for HGT OAG venue and L1-field prediction, at inference time we sample 10 times for target nodes and use the average of predictions to reduce sampling variance of HGT, as we mentioned in footnote 3 on page 6 of our paper.

---

### Decision · Program_Chairs · 2021-01-07
**Final Decision**

**Decision:**

Reject

**Comment:**

This paper proposed an extension of the SIGN model as an efficient and scalable solution to handle prediction problems on heterogeneous graphs with multiple edge types.  The approach is quite simple: (1) sample subsets of edge types, then construct graphs with these subsets of edge types and (2) compute node features on each such graph as if they have only a single edge type, (3) then aggregate the representations from multiple graphs into one using an attention mechanism, and (4) train MLPs on node representations as in SIGN.  Results show that such a simple method can produce quite good results, and is very efficient and scalable.

The reviewers of this paper put it on the borderline, with 3 out of 4 leaning toward rejection.  The most common criticism is the lack of novelty.  Indeed this paper is an extension of prior work SIGN, and the proposed approach is simple.  However, I personally think the simplicity and the great empirical results is rather the strength of this paper.

The authors also did a good job addressing reviewers’ comments and concerns in the discussions, but a few reviewers unfortunately didn’t actively engage in the process.

I'd really encourage the authors to improve and highlight the strength of this paper more and submit to the next venue.